# Eco-Friendly OSN Membranes Based on Alginate Salts with Variable Nanofiltration Properties

**DOI:** 10.3390/membranes13020244

**Published:** 2023-02-17

**Authors:** Evgenia Dmitrieva, Alisa Raeva, Daria Razlataya, Tatyana Anokhina

**Affiliations:** A.V. Topchiev Institute of Petrochemical Synthesis RAS, Leninsky pr. 29, 119991 Moscow, Russia

**Keywords:** organic solvent nanofiltration (OSN), polymeric membranes, natural polymers, sodium alginate, ionic crosslinking, metal cations

## Abstract

In this work, membranes for organic solvents nanofiltration (OSN) based on a natural polymer, sodium alginate, were fabricated. They are chemically stable in organic solvents, including aprotic polar solvents. The unique advantage of these membranes is the absence of toxic reagents and solvents during their production. This ensures the safety and environmental friendliness of the production process. It has been shown that an operation as simple as changing the cation in alginate (Cu^2+^, Fe^3+^, Cr^3+^, Al^3+^, Zn^2+^, Ca^2+^) makes it possible to control the transport and separating properties of membranes, depending on the organic solvent being separated. Therefore, to isolate RemazolBrilliant Blue with MM = 626 g·mol^−1^ from ethanol, membranes based on iron alginate with a rejection *R* = 97% and a permeability of 1.5 kg·m^−2^·h^−1^·bar^−1^ are the most efficient. For isolation of the same solute from DMF and MP, membranes based on calcium alginate with an *R* of about 90% and a permeability of 0.1–0.2 kg·m^−2^·h^−1^·bar^−1^ are the most efficient. The resulting membranes based on natural biodegradable sodium alginate are competitive compared to membranes based on synthetic polymers.

## 1. Introduction

A baromembrane separation process, organic solvent nanofiltration (OSN), is used to concentrate target components/purify organic solvents, replace solvents, or separate solute mixtures [1,2,3]. The advantage of this method over traditional ones (distillation, extraction, etc.) is that it is more energy efficient [4], more economical [5], and environmentally friendly. It ensures the continuity of separation [6]. This allows it to be easily combined with other processes and integrated into flowsheets. Over half a century, hundreds of OSN membranes for solvent regeneration, lubricating oil dewaxing [7], fine organic and pharmaceutical synthesis [8], for example, in peptide synthesis [9], genotoxin removal [10] have been developed.

However, there are still limitations in the use of OSN membranes. The main problem is the tendency of polymers to dissolve, swell, and lose mechanical strength in aggressive aprotic polar solvents [11]. There are only a small number of pure polymers suitable for filtering organic media. Polyether ether ketone (PEEK) is the leader among synthetic polymers. It is resistant to the action of DMF, TEG, alkaline and acid solutions [12]. Chemical [13,14,15,16], thermal [17], UV [18], IR [19] crosslinking are used to improve the chemical stability of most synthetic polymers in organic solvents [20]. P-xylidenediamine [13] and Jeffamine 400 [14] are used forcrosslinking polyimides. Polyamides crosslinked with diisocyanates [15], UV [18] and IR radiation [19] are used to crosslinkthe polysulfone. Dibromoxylin and dibromobutane are used to crosslinkpolybenzimidazole membranes [21].

Membranes can be made from natural polymers [22,23,24,25,26,27]. They can be used as independent membrane materials, as supports for thin selective layers [28], in mixtures with synthetic polymers [29], and also in combinations with each other [30]. Chitosan, cellulose and sodium alginate are insoluble in most organic solvents. This allows them to be used to fabricate OSN membranes. However, it is difficult to select solvents for the preparation of casting solutions. The list of possible solvents for cellulose and chitosan is very limited. Chitosan is soluble in some organic acids [31]. Cellulose is soluble in polar organic solvents such as N,N-dimethyl formamide (DMF), dimethyl sulfoxide (DMSO) and N,N-dimethylacetamide (DMAc) mixed with ethanolamine and/or a suitable inorganic salt [32]. Much attention has beenpaid to the possibilities of creating “green” membranes. They are not only made from natural polymers, but also use non-toxic solvents [33]. Because of this, the use of sodium alginate is more promising. It is known that alginic acid and its derivatives are insoluble in organic solvents [34]. The most important is that sodium alginate is soluble in water [35]. Alginate is an inexpensive polymer that is extracted from brown algae [36] or synthesized by the microorganisms Azotobacter and Pseudomonas [37]. The low cost and unique solubility of alginates make them promising membrane polymers for OSN.

There are many articles in the literature about the membrane application of alginate. The most articles devoted to pervaporation [38,39,40]. Additionally, sodium alginate is known for its use in membrane fuel cells [41] and in ultra- or nanofiltration purification of aqueous solutions [42,43]. Alginates are used as self-supporting membranes, as thin selective layers of composite membranes on mechanically strong supports [44,45], composite membranes with introduced metal oxides [46], or hybrid membranes in combination with other polymers [47,48].

It is known from the literature that the properties of alginate membranes are strongly affected by the structure of the polymer, including chain conformation. This was shown by a team of scientists from Japan [49] using alginate membranes for the pervaporation of water-ethanol mixtures. The polymer conformation was changed by introducing various alkali metal cations that bind to the carboxyl groups of alginic acid. Thus, the authors showed in the series of alginates with cations Li^+^, Na^+^, K^+^, Rb^+^, Cs^+^, swelling in a water-ethanol (10/90%) mixture decreases by almost 2 times, and the separation factor of membranes increases by an order of magnitude from 10^3^ to 10^4^. Thus, a change in the monovalent cation has already led to a strong change in the properties of alginate membranes. An even greater change in membrane properties can be provoked by the variation of metal cations, which differ from each other not only in size, but also in chemistry properties and valency. For example, [50] studied the effect of Ca^2+^, Zn^2+^, Mn^2+^, Co^2+^, Fe^2+^, Al^3+^ cations on the pervaporative separation of mixtures from water and ethanol/isopropanol.

It is known from the literature that the treatment of alginate with salts of polyvalent metals, phosphoric [51], trifluoroacetic acid [52], epichlorohydrin [53], and gluteraldehyde [54] lead to crosslinking of the polymer. It is believed that the crosslinking of alginate with metals occurs in accordance with the egg-crate model of G-guluronic acid residues that form electronegative cavities [55].

In membrane applications, Ca^2+^ [56] is most often used, while Zn^2+^ andFe^3+^ are used less often. For example, Metecan and colleagues created composite membranes on a PAI support by layer-by-layer assembly of PEI and sodium alginate, crosslinking them with transition metal ions [57]. The membranes showed high permeability to aqueous solutions and filtration stability for 72 h. It is known the binding of alginate by ions of polyvalent metals, such as Ba^2+^ and Fe^3+^, lead to the formation of more mechanically strong hydrogels [58]. This may mean an increase in membrane tensile strength and a reduction in the risk of defect formation during long-term filtration.

The study of OSN membranes from crosslinked alginates had not found its researchers for a long time. The first article on this topic appeared as recently as 2020 [59]. In that article, alginate crosslinked with calcium chloride was used as a thin selective layer on cellulose, polyester and crosslinked polyacrylonitrile supports. The article provedthe fundamental possibility of using alginate as a membrane polymer for the purification of alcohols and polar aprotic solvents by nanofiltration. Over the past 2 years, our team of authors has developed this topic. Ref. [26] wasdevoted to the choice of the optimal method for preparation OSN alginate membranes. It was shown that it is preferable to use crosslinking with aqueous salt solutions without the pre-precipitation with organic solvents. In another article [60], it was proved that alginates can be crosslinked with various metal cations. All obtained samples were stable in a wide range of organic solvents. The prospect of further research of alginate salts in OSN was shown in [60].

The purpose of this work is to obtain and study alginate membranes crosslinked with the cations Ca^2+^, Cu^2+^, Fe^3+^, Al^3+^, Cr^3+^, and Zn^2+^. In particular, it is necessary to discuss the process of alginate crosslinking and its effect on the filtration properties of membranes. In addition, the practical application of a variety of crosslinking is important.

## 2. Materials and Methods

### 2.1. Materials

The membranes were made from natural polymer sodium alginate (Rhône-Poulenc, Dijon, France). Distilled water was used as the solvent for sodium alginate. A polyester nonwoven fabric from Crane Technical Materials (Pittsfield, MA, USA) was used as a support. Aqueous solutions of inorganic salts were used as crosslinking agents: CaCl_2_∙2H_2_O, AlCl_3_∙6H_2_O, FeCl_3_∙4H_2_O, ZnCl_2_, CrCl_3_, CuSO_4_∙5H_2_O (HIMMED, Moscow, Russia). The chemically pure organic solvents were used to study the transport properties of membranes: dimethylformamide-DMF (HIMMED, Russia), N-methylpyrrolidone-MP (ACROS, Geel, Belgium), ethanol–EtOH (96% aqueous solution). Negatively charged dyes Remazol Brilliant Blue R (*MW* = 626 g·mol^−1^) and Orange II (*MW* = 350 g·mol^−1^) were used as model solutes.

### 2.2. Characterization of Sodium Alginate and Crosslinked Alginates

The ratio (*M*/*G*) of β-D-mannuronic (*M*) and α-L-guluronic (*G*) acids in sodium alginate was determined by sequential precipitation of a hydrolyzed solution of sodium alginate [61]. It consists of the mild hydrolysis of alginate in a solution of oxalic acid under the microwave radiation with sequential precipitation of *G* and *M* acids at pH = 2.85 and pH = 1.00, respectively.

The molecular weight distribution (MWD) was determined using gel permeation chromatography (GPC) on Agilent 1260MDS (Agilent Technologies, Santa Clara, CA, USA) with refractometric and viscometric detection using two series-connected Polargel (Agilent Technologies, Santa Clara, CA, USA) M 30 × 7.5 columns at an eluent H_2_O flow of 1 mL/min. The volume of the injected sample was 100 µL. Based on the signal of the refractometric and viscometric detectors, the MWD of the samples was calculated using the universal calibration curve according to the Kuhn–Mark–Houwink equation. Standard PEG samples (Agilent Technologies, Santa Clara, CA, USA) with a known molecular weight (EasiVial PEG) were used to construct the calibration curve.

The intrinsic viscosity of sodium alginate was determined using an Ubellode viscometer (Cannon Instrument Company, State College, PA, USA) in a tetraborate buffer with the addition of 0.2 NaCl.

IR spectra of sodium alginate powder were recorded in the ATR reflection mode on a HYPERION-2000 IR microscope (Bruker Corporation, Billerica, MA, USA) coupled to a FS-66 v/s Fourier spectrometer (Bruker Corporation, Billerica, MA, USA): scan 50, Ge crystal, resolution 2 cm^−1^, range 600–4000 cm^−1^. 

The mechanical characteristics were determined on a TT-1100 tensile testing machine (Cheminstruments, West Chester Township, OH, USA) at room temperature (22–24 °C). The traverse movement speed was 3.8 sm·min^−1^. The samples were the pieces of rectangular filmabout 70 mm long and about 10 mm wide. The initial distance between the clamps was 50–60 mm. The stresses were calculated for the initial section of the sample.

The morphology of the samples of supports and composite membranes (surface and transverse cleavages) was studied using a scanning electron microscope Phenom XL G2 Desktop SEM (Thermo Fisher Scientific, Waltham, MA, USA), equipped with a module for energy-dispersive elemental spectroscopy (EDX). Using a magnetron sputter Cressington 108 auto Sputter Coater (Ted Pella, Redding, CA, USA) a thin layer of gold with a thickness of 5–10 nm was applied to the surface of the samples. The value of the accelerating voltage during the measurement was 15 keV.

### 2.3. Preparation of Composite Membranes from Crosslinked Alginate

The composite membranes were fabricated by the deposition of a toplayer of 10 wt.% solution of sodium alginate in water on the non-woven porous support made of polyester nonwoven by using the doctor blade with a gap thickness of 200 µm (Figure 1). The membrane was placed in the precipitation bath (0.35 mol-eq·L^−1^ aqueous solutions of inorganic salts) for 30 min. This method of preparingalginate membranes was recognized as optimal based on the results of our previous work [26].

After crosslinking, the membranes were washed twice with distilled water for 10 min, after which they were kept in water until nanofiltration tests. To perform strength tests and by taking SEM photographs, the membranes were dried. The selective layer was removed from the support.

### 2.4. Research of Nanofiltration Properties

The composite membranes based on crosslinked alginate were tested in nanofiltration of organic solutions in dead-end cells at 10 bar. The mixture in the cells was constantly stirred with magnetic stirrers. The permeate flow was determined by the gravimetric method. A liquid receiver was installed at the outlet of the cell. The mass of permeate passing through the membrane during the experiment was measured on a Sartorius laboratory balance with a measurement error of 0.001 g. Membrane performance was characterized by liquid permeation (*P*) and was calculated by Equation (1) [62]:(1)$$P=\frac{m}{S\times \Delta t\times \Delta p}$$
where *m* is the weight of permeate (kg) passed through a membrane with an area *S* (m^2^) over a time Δ*t* (h), Δ*p* is the pressure drop.

The optical density of solutions was measured with PE-5400UF spectrophotometer (PromEcoLab, Saint Petersburg, Russia). The concentrations of model compounds (dyes) in the feed and permeate were determined using the calibration curve. The rejection *R* (%) was calculated and used to evaluate the separation characteristics [62,63] of the membrane (2):(2)$$R=\left(1-\frac{C_{p}}{C_{0}}\right)\times 100\%$$
where *C*_p_ and *C*_0_ are the dye concentrations in the permeate and feed, respectively.

## 3. Results

### 3.1. Study of the Original Polymer—Sodium Alginate

An important characteristic of sodium alginate is the ratio of mannuronic and guluronic acid residues, which determines the tendency of the polymer to gel. A direct determination of the ratio of acid residues was carried out by the precipitation method [61], which gave a numerical value (*M*/*G*) = 5.7. A significant predominance of mannuronic acids over guluronic acids is confirmed indirectly by the IR spectrum of sodium alginate (Figure 2). The intensity of the absorption band of stretching vibrations at 819 cm^−1^ (M) is much higher than the intensity of the band at 782 cm^−1^ (G).

Gel permeation chromatography was used to determine the molecular weight of the polymer, equal to *M*_w_ = 1.2∙10^6^. The polydispersity index was *M*_w_/*M*_n_ = 6.6, which may indicate the presence of polymers with molecular weights of a wide range in the sample [64].

The intrinsic viscosity, measured on an Ubbelohde viscometer, of the sodium alginate was 6.87 (cm^2^∙g^−1^). The viscosity of the 10% aqueous solution of sodium alginate used for membrane casting was 245,000 mPa∙s.

### 3.2. Study of the Alginate Crosslinking

It is known from the literature the addition of divalent and trivalent metal cations to alginate lead to ionic crosslinking [65]. At the same time, it is assumed that due to the folded conformation of *L*-guluronate, “cells” are formed in the polymer, inside which there are metal cations [66] (Figure 3). Only *L*-guluronic acids are involved in the crosslinking of the polymer.

The IR spectrum of all crosslinked alginates well conveys the structure of alginic acid salts. The most intense bands in the region of 1595, 1425 cm^−1^ correspond to antisymmetric and symmetric vibrations of the C = O bond in carboxylate ions, the position of the maximum of which and the splitting depend on the nature of the cation. The intense split band at 1030 cm^−1^ refers to vibrations of the C-OH bond in pyranose rings and glycosidic bonds C-O-C, the bands on the short-wavelength shoulder at 1079 and 1128 cm^−1^ indicate inhomogeneity in optical isomers. Very weak bands in the region of 1730–1740 cm^−1^ from the acidic form of the carboxyl group, the content of such groups in all samples is negligible. The characteristic shape of the spectrum does not change, since crosslinking of alginate with metals involves carboxyl and hydroxyl groups, forming ionic and coordination bonds, without causing a change in the chemical structure of the polymer and the formation/disappearance of functional groups. However, the spectrum narrows in the region of 3350 cm^−1^ vibrations (Figure 4), which, according to the literature data, is associated with the formation of hydrogen bonds between carboxyl groups –OH and metal cations [67], which are necessary for the formation of “cells” from *L* residues—guluronic acids.

The crosslinking of the polymer was carried out usingthe phase inversion method, in which the open surface of the alginate had the greatest contact with salt solutions. In this case, an uneven distribution of metals and the formation of a selective layer with a metal concentration gradient over its thickness can be assumed.

However, a high concentration of metals in alginates, ω(Me^n+^) = 15–25 wt.% (Table 1), indicates crosslinking occurs not only at the surface, but also along the thickness. EDX data for metal alginates confirm this (Figure 5).

In Figure 5, the corresponding crosslinking cations are color-coded, andare evenly distributed throughout the thickness of the crosslinked alginates. It should be noted that the concentrations of sodium in all obtained samples are 0% wt. (Table 1). The crosslinking is complete. Multivalent metal cations, penetrating into the alginate structure, form bonds between polymer chains (Figure 6).

In this case, the resulting multidimensional structure of the crosslinked polymer strongly depends on the nature of the cation. This is due to the different valency of cations (Figure 6), as well as their different sizes (Table 2).

The valency of the metal affects the branching of the resulting structure, and the size of the cation affects the distance between the crosslinked alginate chains. All these must lead to a change in the operational and filtration properties of alginate membranes.

In the course of the crosslinking study, it was found that sodium alginate is crosslinked by di- and trivalent metal cations by an ionic mechanism. In this case, bonds between positively charged cations and negatively charged carboxyl groups of the polymer are formed. In addition, coordination bonds with the hydroxyl groups of alginate are formed. This is confirmed by IR spectroscopy. Crosslinking of the polymer occurs not only on the surface, but throughout its entire thickness, which is clearly shown by the EDX data. Such crosslinking leads to high concentrations of metals in the membrane matrix: 15–25% wt. The cations used in this work differ from each other in size and valency. This, combined with their high concentration, leads to a strong change in the structure of the crosslinked polymer.

### 3.3. Effect of Metal Cations on the Mechanical and Nanofiltration Properties of Membranes Based on Crosslinked Alginates

Alginates crosslinked covalently with gluteraldehyde or ionic calcium cations are insoluble in alcohols [38,39] and other organic solvents such as 1,4-dioxane, tetrahydrofuran [68], DMF, and MP [59]. Studies have shown that aluminum, zinc, iron (III), chromium (III) and copper alginates are stable in protic (ethanol) and aprotic polar solvents (MP, DMF). The membranes do not dissolve in organic solvents when stored for more than a month, and do not show a mass deficit. This allows them to be used for nanofiltration of the corresponding organic solvents.

The change in the crosslinking cation makes it possible to change the permeation of membranes for pure organic solvents in a wide range (Table 3). Thus, in the series of alginates Cr^3+^-Cu^2+^/Ca^2+^-Al^3+^-Fe^3+^-Zn^2+^, the permeation ofprotic and aprotic polar solvents increases by 23–55 times. Note that calcium alginate is more permeable in the case of polar ethanol filtration than copper alginate. Thispattern is reversed in the case of filtering MP and DMF. At the same time, the aprotic solvent DMF in all cases is characterized by a higher permeability compared to MP. This may be due to the lower molecular weight (73 < 99 g·mol^−1^) and viscosity (0.92 < 1.79 mPa·s). In addition, the variation of the cation makes it possible to change the mechanical strength of the formed films over a wide range, more than 15 times—from 6 to 97 MPa.

The rejection properties of the membranes were studied using the model dye Remazol Brilliant Blue R (*MW* = 626) with negative charge. Rejections of other types of dyes are shown in Appendix A. The permeation of the mixtures was in all cases lower than for pure solvents (Table 4).

The rejection varied in the ranges of 57–98, 27–90, 11–93% for solutions of Remazol Brilliant Blue R in ethanol, DMF and MP, respectively (Table 4). The optimal crosslinking cation could be chosen for each of the investigated solvents. For example, membranes with the best combination of transport and separating properties in ethanol are crosslinked with Al^3+^ and Fe^3+^, in DMF and MP-crosslinked with Al^3+^ and Ca^2+^. For all solvents, membranes crosslinked with Cr^3+^ cations demonstrated the highest rejection coefficients, at 90–98%, but they hadthe lowest permeation, 0.01 kg·m^−2^·h^−1^·bar^−1^. Another crosslinking cation, for example, Zn^2+^, is ineffective in the purification of solvents from solutes with *MW* = 626 g·mol^−1^. However, it can allow very fast pre-roughing of ethanol at *R* = 57% or can be used to clean organic solvents from higher molecular weight contaminants. Thus, changing the crosslinking cation makes it possible to obtain membranes for various filtration tasks based on the same initial polymer, sodium alginate.

Figure 7 shows that the permeability of the membranes drops sharply during the first filtration time. Stabilization of permeability for ethanol occurs much earlier than for DMF. In the first case, the permeability reaches 0.45 kg·m^−2^·h^−1^·bar^−1^ after one hour of filtration. After that, the permeability practically ceases to change, and decreases after 24 h of filtration by 10% to 0.4 kg·m^−2^·h^−1^·bar^−1^. Permeability stabilization in DMF occurs only after 4 h of filtration, reaching a permeability value of 0.2 kg·m^−2^·h^−1^·bar^−1^. Despite different permeability stabilization times, further changes are insignificant in both ethanol and DMF. The rejection also changes over time. The *R* for Remazol Brilliant Blue R decreases after 24 h of filtration from 96 to 82% in DMF, and from 71 to 58% in ethanol. The change in filtration characteristics over time is not accompanied by a change in membrane morphology, as shown in Appendix A.

Changing the crosslinker allows not only to vary the permeability and rejection, but also to vary the cut-off molecular weight (MWCO). Based on the experimental results (Table 5) it can be stated that cut-off molecular weight for zinc and calcium alginate MWCO ≥ 626, iron, aluminum and copperalginate MWCO ≤ 626, chromium alginate MWCO ≤ 350.

Alginate-based membranes can be used to separate solutes with different molecular weights. To do this, the membrane must have a high rejection for the higher molecular weight solute and a low rejection for the lower-molecular-weight solute. The iron alginate-based membrane satisfies this condition (Table 5). It rejects Remazol Brilliant Blue R (*MW* = 626 g·mol^−1^) from ethanol by 97% and Orange II (*MW* = 350 g·mol^−1^) by only 10% (Table 5). Such a difference in rejection makes it possible to separate the mixture of these dyes. The purple solution (Remazol Brilliant Blue R + Orange II) turns to orange permeate (Orange II) whenfilteredthrough iron alginate (Figure 8).

The conducted studies confirm the earlier assumption. Indeed, a change in the crosslinking cation and a corresponding change in the structure also lead to a variation in the nanofiltration and mechanical properties of alginate membranes. Therefore, the permeability for pure solvents with varying cations changes in the range of 23–55 times, and mechanical strengthmore than 15 times. The membrane selectivity also varies. The rejection is 57–98, 27–90, 11–93% for solutions of Remazol Brilliant Blue R (*MW* = 626) in ethanol, DMF and MP, respectively. The rejection also varies for the low-molecular-weight dye Orange II (*MW* = 350). All these facts make it possible to use the obtained alginate salts to solve various problems of separation, including the fractionation of dyes.

### 3.4. Comparison of Properties of Membranes Based on Crosslinked Alginates with Membranes Based on Synthetic Polymers

The use of the chelating properties of metal cations in the preparation of membranes has already been studied in the literature to change the properties of supports [69] and selective layers [70,71]. For example, it was shown in [72] that this approach makes it possible to obtain much thinner and more permeable OSN polyamide membranes.In another work [73], crosslinking of support with calcium cations is used to increase the resistance of membranes to organic solvents and increase their permeability. The article [74] shows the possibility of using copper (I) to fine tune the molecular sieving performance of polybenzimidazole membranes. The present work shows not only the positive effect of chelating cations on the properties of OSN membranes, but also demonstrates the possibility of a significant change in the filtration properties of membranes from the same polymer by replacing the chelating metal. The membranes obtained in this work are competitive compared to membranes based on synthetic polymers (Table 6).

Alginate membranes are not inferior in terms of transport and rejection properties to membranes from synthetic polymers (PAN, nylon, polyimines, etc.). This fact is very important. It makes it possibleto replace polymers and methods of preparing membranes from synthetic and often toxic to natural and safe for humans and the environment.

## 4. Conclusions

Membranes based on water-soluble sodium alginate were prepared using an environmentally friendly method. The alginate membranes are stable in organic solvents. This allows them to be successfully used in OSN processes.

It was shown that not only calcium chloride, but also salts of other polyvalent metals can be used for membrane crosslinking. An operation as simple as changing the crosslinking solution significantly affects the properties of membranes. The mechanical tensile strength changed by more than 18 times and increased for Cr–Cu–Zn–Fe–Al–Ca alginates from 6 to 111 MPa. The change in the crosslinking cation makes it possible to vary the transport and separating properties of membranes in a wide range: by 56 and 9 times, respectively. This makes it possible to use the alginate membranes for various separation tasks. Zinc alginate can be used for coarse purification. Chromium alginate can be used for the fine purification of solvents. In addition, an iron alginatecan be used to separate solutes with different molecular weights (*MW* ≥ 626 and *MW* ≤ 350).

Alginate salts are promising in OSN for the purification of polar protic and aprotic solvents. In addition, they are competitive with membranes based on synthetic polymers. Alginate membranes purify organic solvents with an efficiency of more than 80% with permeabilities comparable to those known in the literature.

The next step in the development of this work is to establish the mechanisms of permeability of the developed membranes. This will provide clues as to why the crosslinking cation is so important and allows for such a wide variety of properties in alginate membranes.

## Figures and Tables

**Figure 1 membranes-13-00244-f001:**
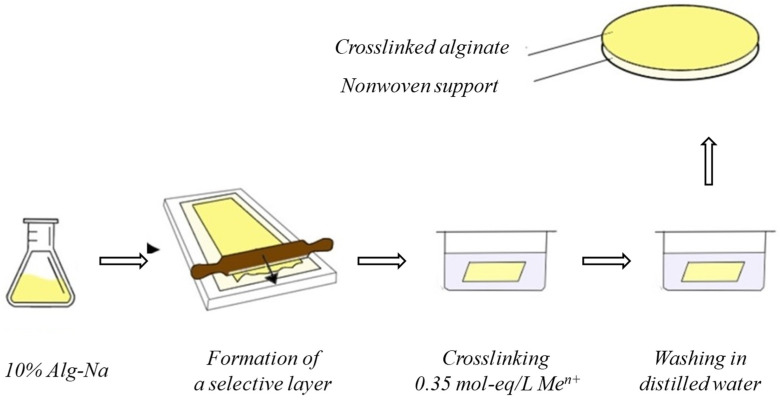
Scheme of preparation of composite membranes from crosslinked alginate.

**Figure 2 membranes-13-00244-f002:**
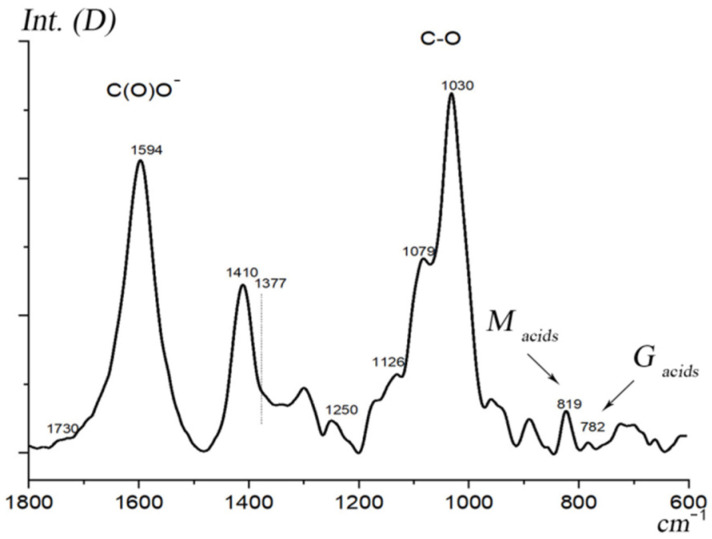
Mannuronic (*M*) and guluronic (*G*) acids in the IR spectrum of sodium alginate.

**Figure 3 membranes-13-00244-f003:**
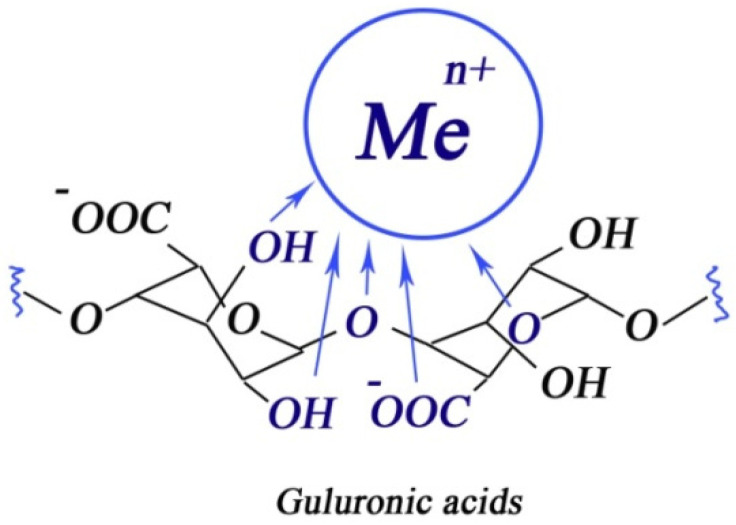
Proposed mechanism of alginate crosslinking by polyvalent metals.

**Figure 4 membranes-13-00244-f004:**
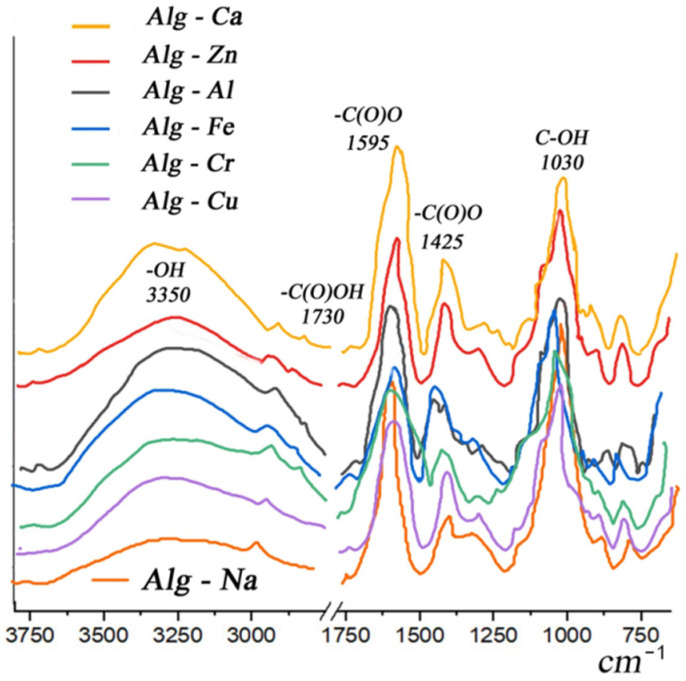
IR spectra of polyvalent metal alginates.

**Figure 5 membranes-13-00244-f005:**
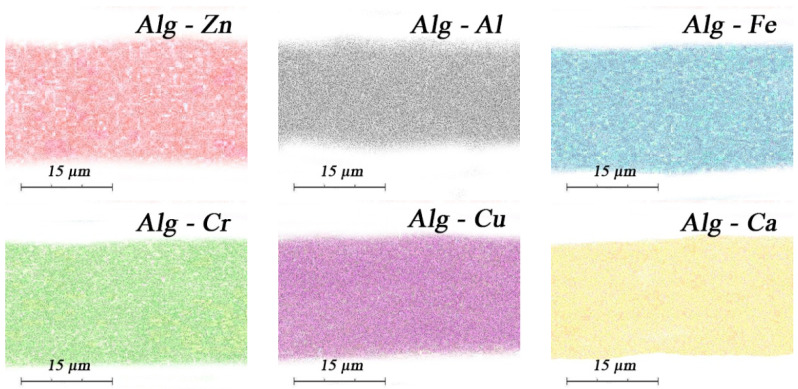
Crosslinking of alginates throughout the film thickness (EDX data).

**Figure 6 membranes-13-00244-f006:**
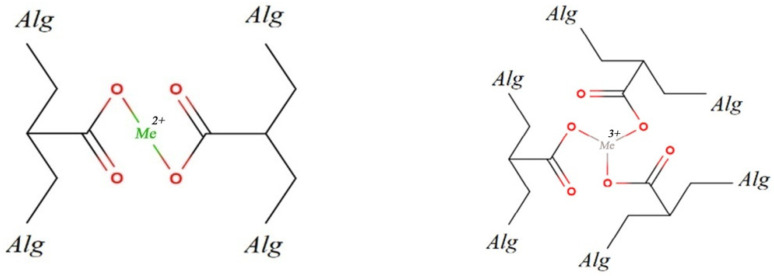
Scheme of crosslinking alginate (*Alg*) with metals of different valence (Me^n+)^.

**Figure 7 membranes-13-00244-f007:**
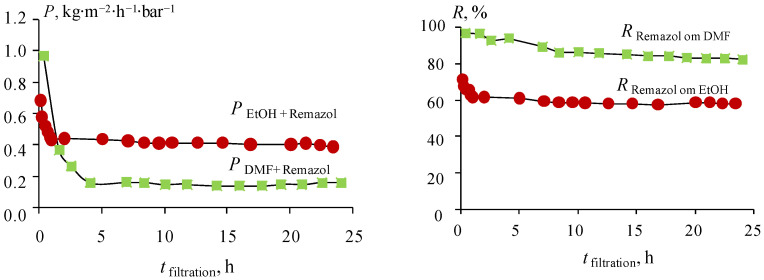
Changes in the nanofiltration characteristics of a membrane based on calcium alginate with time of filtration.

**Figure 8 membranes-13-00244-f008:**
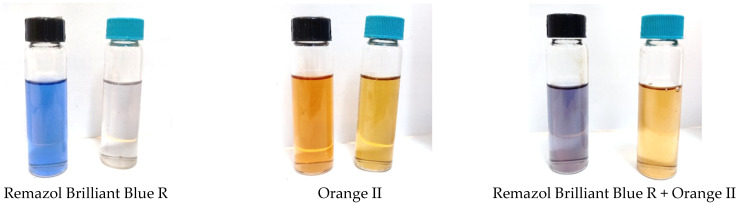
Comparison of the color of permeate during filtration of various dyes and their mixture in ethanol through membranes based on iron (III) alginate.

**Table 1 membranes-13-00244-t001:** Comparison of the elemental composition of films from alginates of various metals.

Me^n+^	Al^3+^	Ca^2+^	Fe^3+^	Zn^2+^	Cu^2+^	Cr^3+^
ω(Me^n+^), mol. %	0.55	0.46	0.44	0.37	0.39	0.32
ω(Me^n+^), wt. %	15.0	18.4	24.9	24.0	25.0	16.7
ω(Na^+^), wt. %	0	0	0	0	0	0

ω(Me^n+^), mol.—molar concentration of the crosslinking cation in the polymer matrix (%), ω(Me^n+^), wt.—weight concentration of the crosslinking cation in the polymer matrix (%).

**Table 2 membranes-13-00244-t002:** Sizes of atoms and ions of crosslinking metals.

Me^n+^	Zn^2+^	Cu^2+^	Ca^2+^	Al^3+^	Fe^3+^	Cr^3+^
*R*_atom_, pm	142	145	197	118	156	166
*R*_ion_, pm	83	96	99	57	63	64

*R*_atom_, *R*_ion_—are the radius of the crosslinking metal atom and ion (pm), respectively.

**Table 3 membranes-13-00244-t003:** The effect of a crosslinking cation on the mechanical strength and permeation of alginate membranes for pure solvents.

Me^n+^	*σ*_B_, MPa	*P* _EtOH_	*P* _DMF_	*P* _MP_
Cr^3+^	6	0.2	0.1	0.1
Cu^2+^	23	1.3	1.1	0.9
Ca^2+^	111	2.3	1.0	0.4
Al^3+^	97	2.6	2.1	0.8
Fe^3+^	62	3.9	4.4	1.3
Zn^2+^	49	7.6	5.5	2.3

*σ*_B_—tensile strength (MPa); *P*_EtOH_, *P*_DMF_, *P*_MP_—membrane permeation of ethanol, DMF and MP, respectively (kg·m^−2^·h^−1^·bar^−1^).

**Table 4 membranes-13-00244-t004:** The effect of a crosslinking cation on the nanofiltration properties of alginate membranes according to RemazolBrilliant Blue R model solutions in organic solvents.

Me^n+^	*P*_EtOH + R_,	*R*_EtOH + R_, %	*P* _DMF + R_	*R*_DMF + R_, %	*P* _MP + R_	*R*_MP + R_, %
Cr^3+^	0.01	98	0.01	90	0.01	90
Cu^2+^	1.50	94	0.40	83	0.60	51
Ca^2+^	0.45	65	0.20	87	0.10	93
Al^3+^	0.50	97	0.40	83	0.20	83
Fe^3+^	1.50	97	2.20	63	2.00	14
Zn^2+^	5.60	57	2.00	27	0.70	11

*P*_EtOH + R_, *P*_DMF + R_, *P*_MP + R_—membrane permeationof Remazol Brilliant Blue R solutions in ethanol, DMF and MP, respectively (kg·m^−2^·h^−1^·bar^−1^); *R*—RemazolBrilliant Blue rejection coefficient R (*MW* = 626 g·mol^−1^) (%).

**Table 5 membranes-13-00244-t005:** The influence of a crosslinking cation on the cut-off molecular weight of alginate membranes.

Me^n+^	*R*_RemazolBrilliantBlueR_ (*MW* = 626)	*R*_OrangeII_ (*MW* = 350)
Cr^3+^	98%	90%
Fe^3+^	97%	10%
Al^3+^	97%	13%
Zn^2+^	57%	8%
Ca^2+^	65%	10%
Cu^2+^	94%	47%

*R*_RemazolBrilliant Blue R_—rejection coefficient of RemazolBrilliantBlueR from ethanol (%), *R*_OrangeII_—rejection of OrangeII from ethanol (%).

**Table 6 membranes-13-00244-t006:** Comparison of nanofiltration properties of the developed membranes with literature data.

Polymer	Solute (*MW*)	*P* _solvent + solute_	*R*	Ref.
Solvent–EtOH				
GO + Pebax/PAN	Brilliant Blue R (826)	1.90	95	[75]
**Alginate-Fe**	**Remazol Brilliant Blue R (626)**	**1.50**	**97**	**The present work**
GNPs/CA	Bromothymol blue (624)	1.50	82	[75]
POSS + Catechol/PI	Rose Bengal (1017)	1.26	99	[75]
(GO + PEI)/ PAN	PEG (200)	1.10	97	[75]
MIL-53 (Al)/PMIA	Brilliant Blue G (854)	0.70	94	[75]
**Alginate-Al**	**Remazol Brilliant Blue R (626)**	**0.50**	**97**	**The present work**
Solvent–DMF				
**Alginate-Fe**	**Remazol Brilliant Blue R (626)**	**2.20**	**63**	**The present work**
PA/crosslinked P84 PI	Styrene oligomers (236)	1.50	91	[76]
**Alginate-Al**	**Remazol Brilliant Blue R (626)**	**0.40**	**83**	**The present work**
PAES	Styrene oligomer (1595)	0.37	99	[77]
Alginate-Ca	Vitamin B12 (1355)	0.23	70	[59]
(PDDA/SPEEK)/ (PAN-H/Si)	Rose Bengal (1017)	0.07	89	[76]
PPy/PAN-H	Rose Bengal (1017)	0.05	91	[76]
(PS-b-PEO/PAA) /alumina	Ethylene glycol Oligomers (420)	0.02	78	[76]
**Alginate-Cr**	**Remazol Brilliant Blue R (626)**	**0.01**	**90**	**The present work**
Solvent–MP				
Multilayer GO/nylon	Methyl Orange (327)	0.80	99	[75]
**Alginate-Al**	**Remazol Brilliant Blue R (626)**	**0.20**	**83**	**The present work**
Alginate-Ca	Vitamin B12 (1355)	0.10	80	[59]

*P*_solvent + solute_—permeation of membranes for a solution of a dissolved substance (solute) in an organic solvent (kg·m^−2^·h^−1^·bar^−1^); *R*—rejection coefficient of a solute (%).

## Data Availability

Not applicable.

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
