# Peer review of "Eco-Friendly OSN Membranes Based on Alginate Salts with Variable Nanofiltration Properties"

_membranes, 2023, doi:10.3390/membranes13020244_

Round 1

Reviewer 1 Report

The article entitled “Membranes based on alginate salts with variable nanofiltration properties” is interesting and fits in principle with the scope of “Membrane” journal. However, there are some points which authors must work on. Detailed comments are shown below:

The manuscript needs extensive editing of English language. Many sentences are unclear and poor.

In abstract, OSN should be defined at first mention

Introduction:

1st line: replace “organic nanofiltration”  with “organic solvent nanofiltration”

Provide reference(s) for use of OSN membranes for solvent recovery.

“PEEK polymer” should be defined at first mention

Page 1, Ln 37 “The use of most other polymers usually involves chemical, thermal, UV, IR crosslinking or annealing”. Clarify and provide relevant reference.

“So, polyimides are сrosslinkedp-xylidene diamine 40 [12] or Jeffamine 400 [13], polyamides - with diisocyanates [14], polysulfones - with UV 41 [15] and IR radiation [16], polybenzimidazole membranes - with dibromoxylin or di-42 bromobutane [17].” Unclear. Please rewrite it.

“There are a number of natural polymers that are stable in organic solvents: cellulose [18], chitosan [19,20], sodium alginate [21-23].” this sentence is not accurate. Chitosan has very limited solubility in organic solvent. Sodium alginate does not dissolve in organic solvents. Please rewrite it.

Ln 81 “AuseMetecan” revise.

In Figure 1, SA abbreviation should be identified. You did not use it in the whole manuscript. Sodium alginate was coded as Alg-Na in Fig. 4. Please standardize.

Provide adequate references for equations 1 and 2.

Ln 179 “The polydispersity index was Mn/Mw=6.6” wrong. PDI is defined by Mw/Mn.

Ln 264 “rejeaction” typos.

Author Response

Reviewer 1

Thank you very much for your review and comments. All of comments were taken into account. Corresponding corrections have been made in the manuscript. All corrections are highlighted in yellow.

Responses to comments can be read below.

  1. In abstract, OSN should be defined at first mention

We have given a definition of the abbreviation OSN in the abstract (page 1)

«In this work, membranes for organic solvent nanofiltration (OSN) were fabricated based on a natural polymer sodium alginate»

  1. 1st line: replace “organic nanofiltration” with “organic solvent nanofiltration”

The phrase “organic nanofiltration” has been corrected to “organic solvent nanofiltration” (page 1)

  1. Provide reference(s) for use of OSN membranes for solvent recovery.

References for use of OSN membranes for solvent recovery (doi.org/10.1039/C2GC35216H, doi.org/10.1021/sc5004083) are given in the first sentence of the introduction (page 1)

  1. “PEEK polymer” should be defined at first mention

The definition of the abbreviation PEEK polymer (polyetheretherketone) has been added to the text of the manuscript (page 1)

  1. Page 1, Ln 37 “The use of most other polymers usually involves chemical, thermal, UV, IR crosslinking or annealing”. Clarify and provide relevant reference.

References have been added for each type of polymer crosslinking. Some of them have already been mentioned in the manuscript. In addition, new references have been added: doi.org/10.1016/j.progpolymsci.2010.08.003, doi.org/10.1016/j.progpolymsci.2012.11.001 (page 1)

  1. “So, polyimides are сrosslinked p-xylidene diamine 40 [12] or Jeffamine 400 [13], polyamides - with diisocyanates [14], polysulfones - with UV 41 [15] and IR radiation [16], polybenzimidazole membranes - with dibromoxylin or di-42 bromobutane [17].” Unclear. Please rewrite it.

This sentence has been rewritten (page 1)

«p-xylidene diamine [12] and Jeffamine 400 [13] are used to crosslinking polyimides. Polyamides are crosslinked with diisocyanates [14], UV [15] and IR radiation [16] are used to crosslinking the polysulfone.Dibromoxylin and dibromobutane are used to crosslinking polybenzimidazole membranes [17]».

  1. “There are a number of natural polymers that are stable in organic solvents: cellulose [18], chitosan [19,20], sodium alginate [21-23].” this sentence is not accurate. Chitosan has very limited solubility in organic solvent. Sodium alginate does not dissolve in organic solvents. Please rewrite it.

Thanks for your valuable comment. Information about the stability of natural polymers in organic solvents has been corrected (page 2). Chitosan is soluble in organic acids. Information about this has been added to the manuscript(doi.org/10.1111/j.1365-2621.2002.tb11382.x).

«Membranes can be made from natural polymers [18–23]. Chitosan, cellulose and sodium alginate are insoluble in most organic solvents. This allows them to be used to fabricate OSN membranes. However, it is difficult to select solvents for the preparation of casting solutions. The list of possible solvents for cellulose and chitosan is very limited. Chitosan is soluble in some organic acids [Park, 2002]. Cellulose is soluble in polar organic solvents N,N-dimethylformamide (DMF), dimethyl sulfoxide (DMSO), N,N-dimethylacetamide (DMAc) mixed with ethanolamine and/or a suitable inorganic salt [24]. The use of sodium alginate is more promising. It is known alginic acid and its derivatives are insoluble in organic solvents [27]. It is easy to prepare casting solutions from this polymer because it is soluble in water [26]. Alginate is an inexpensive polymer that is extracted from brown algae [28] or synthesized by the microorganisms Azotobacter and Pseudomonas [29]. The low cost and unique solubility of alginates make them promising membrane polymers for OSN».

  1. Ln 81 “AuseMetecan” revise

A space have been added between the first and last name (page 3)

  1. In Figure 1, SA abbreviation should be identified. You did not use it in the whole manuscript. Sodium alginate was coded as Alg-Na in Fig. 4. Please standardize.

Sodium alginate is coded as "Alg-Na" in the manuscript. The code in figure 1 has been corrected (page 5)

  1. Provide adequate references for equations 1 and 2

Links to equations 1 and 2 have been are added to the text of the manuscript (page 5). In addition, references have been added to articles that use the same equations (10.1016/S1383-5866(02)00095-3, 10.1039/C9TA12258C).

  1. Ln 179 “The polydispersity index was Mn/Mw=6.6” wrong. PDI is defined by Mw/Mn.

Thanks for correction. There was a typo in the equation in the article. The calculations were correct. Mw=1.2∙106, Mn=1.8∙105, Mw/Mn = 6.6. The equation in the text of the manuscript has been corrected (page 6)

  1. Ln 264 “rejeaction” typos.

The typo has been corrected (page 9).

Reviewer 2 Report

1. The adsorption of the dyes solutes should be decoupled from the rejection on the membranes. Mass balances need to be closed. Report the mass balances in a supporting table: mass/moles of solutes in the feed, permeate and retentate. What is adsorbed on the membrane?

2. Photographs, ATR-FTIR and SEM of the membranes before versus after dye filtration should be reported to show any possible changes. The membranes are not sufficiently characterized.

3. It is promising that the membranes showed good performance in DMF, which is a difficult solvent to filter. Long term filtration performance in continuous mode (few days or a week) should be shown to demonstrate if there is any change in the separation performance including rejection and permeation.

4. Non-charged non-dye solutes should be filtered as well and rejection reported. Preferably a curoff curve should be shown.

5. The authors take advantage of the metal ion chelating properties of the polymer to obtain OSN membranes. Similar approaches have been published already with various polymers by various groups inc. Jian Jin, Yan Wang, Szekely, Peinemann groups, which should be discussed, and the approaches and performance of these metal ion/polymer pairs compared in a final table before the Conclusions (i.e. Table 7).

6. The novelty of the research needs to be clarified in light of the similar research on alginate membranes for OSN that the authors published recently, one cited in the submitted manuscript but the other one is not mentioned (10.3390/membranes12020235; 10.4028/www.scientific.net/KEM.899.745).

7. Natural/green polymers for OSN are emerging and recent achievements should be mentioned (10.1016/j.memsci.2018.09.042; 10.1039/C8TA02697A; 10.1016/j.memsci.2022.120743; 10.1016/j.advmem.2022.100041).

8. The degree of crosslinking should be determined and reported for all membranes studied.

9. Can the developed membranes be used for catalytic purposes? Some of the metal-polymer chelated systems are excellent catalysts. Is this a potential application area (catalytic membrane reactors) for these membranes?

10. The authors propose an OSN cascade but it is out of the blue in the manuscript. Some discussions on the application of cascades in OSN should be added, the topic introduced. Separation performance should be evaluated experimentally or at least the rejections should be used for some cascade modeling to demonstrate the applicability and potential of the proposed cascade configuration.

Author Response

Reviewer 2

Thank you very much for your review and careful reading of our work. Your comments allowed us to finalize the manuscript and significantly improve it. Corresponding corrections have been made to the manuscript. All corrections are highlighted in purple.

Responses to comments can be read below. Note that the answer to comments 1 and 4 is general, because questions are interrelated with each other.

  1. The adsorption of the dyes solutes should be decoupled from the rejection on the membranes. Mass balances need to be closed. Report the mass balances in a supporting table: mass/moles of solutes in the feed, permeate and retentate. What is adsorbed on the membrane?
  2. Non-charged non-dye solutes should be filtered as well and rejection reported. Preferably a cut-off curve should be shown.

Emphasis is placed on negatively charged model substances in the study of the separating properties of alginate membranes. This is due to the experimental facts obtained in our study (Table 1).

Table 1. Rejection (R, %) of model substances of different charge and molecular weight for a composite membrane with a selective layer of calcium alginate (Alg-Ca)

Model substance

Charge

MW, g‧mol-1

RDMF

REtOH

Solvent Blue 35

Neutral

350

53

12

Orange (II)

Negative

350

55

10

Remazol Brilliant Blue R

Negative

626

87

65

Safranine O

Positive

350

18

8

Victoria Blue B

Positive

506

30

7

Table 1 presents rejection different types of dyes for a composite membrane with a selective layer of calcium alginate (Alg-Ca). It has been shown the rejection capacity of alginate membranes is low for positively charged substances. Perhaps this is due to the sorption of these substances into the polymer matrix. Sorption occurs even when the membranes are kept in organic solutions without pressure (Figure 1). At the same time, alginate is characterized by the absence of sorption of negatively charged and neutral substances, even when filtered under pressure (Figure 1).

Safranine O

Victoria Blue B

Solvent Blue 35

Remazol Brilliant Blue R

Orange (II)

Figure 1 – sorption of dyes by Alg-Ca from ethanol

The membrane based on calcium alginate has approximately the same rejection of non-sorbing substances (table 1). R of negative Orange (II) and neutral Solvent Blue 35 with the same molecular weight (350 g‧mol-1) is almost the same and amounts R = 53-55% from DMF and R = 10-12% from ethanol.

This information is available in additional materials to the manuscript. In the Supplementary 1, readers can find the rejection coefficients of different types of substances for alginate membranes (question 4). Presentation of cut-off curves is currently not possible due to the small data set for the substances with the various molecular weights. This is a prospect for further research. The absence of sorption of negatively charged dyes Orange (II) and Remazol Brilliant Blue R makes it possible not to indicate the mass balances of substances (question 1).

  1. Photographs, ATR-FTIR and SEM of the membranes before versus after dye filtration should be reported to show any possible changes. The membranes are not sufficiently characterized.

Alginate membranes are known to be dense with no visible porosity (doi.org/10.1016/j.memsci.2019.117615). The same fact was revealed in this work and shown in previous ones (10.3390/membranes12020235; 10.4028/www.scientific.net/KEM.899.745). In this regard, mechanical deformation and changes under the influence of pressure do not occur during filtration. SEM photographs do not show any changes before and after filtration (Figure 2).

Before filtration

After filtration

Figure 2. SEM images of calcium alginate before and after filtration the solution
EtOH + Remazol Brilliant Blue R

As has been shown earlier, filtration of solutions of negatively charged dyes in organic solvents does not lead to adsorption and visible changes in membranes (Figure 1). The same fact was confirmed by holding the membranes in dye solutions in organic solvents. The optical density of the solutions did not change when the membranes were stored in them. There is no leaching of metals from the matrix of membranes during filtration. Thus, according to the EDX data, the concentration of iron before filtration is 24.9%, after filtration is 25.0%.

Thus, it can be argued alginate membranes are stable and are not subject to significant changes during OSN. This information is available in the Supplementary 2.

  1. It is promising that the membranes showed good performance in DMF, which is a difficult solvent to filter. Long term filtration performance in continuous mode (few days or a week) should be shown to demonstrate if there is any change in the separation performance including rejection and permeation.

Membranes based on chromium and calcium alginate have the highest selectivity in DMF. Chromium alginate has a much lower permeability and is less promising for the filtration process. In this regard, long-term filtration of a DMF solution has been demonstrated using calcium alginate (figure 3 in this text, figure 7 in the new manuscript). In a previous version of the manuscript, the change in permeability and rejection coefficients over time was characterized in ethanol. The stability of the characteristics depends little on the nature of the solvent. To show this, the new illustration contains data for ethanol and DMF. Figure 7 in the manuscript has been changed to a new one (page 10). The description for it has also been changed (page 10).

Figure 3. Changes in the nanofiltration characteristics of a membrane based on calcium alginate with time of filtration

The figure 3 (figure 7 in the manuscript) shows the permeability of the membranes drops sharply during the first filtration time. Stabilization of permeability for ethanol occurs much earlier than for DMF. In the first case, the permeability reaches 0.45 kg‧m-2‧h-1‧bar-1 after 1 hour of filtration. After that, the permeability practically ceases to change and decreases after 24 hours of filtration by 10% to 0.4 kg‧m-2‧h-1‧bar-1. Permeability stabilization in DMF occurs only after 4 hours of filtration, reaching a permeability value of 0.2 kg‧m-2‧h-1‧bar-1. Despite different permeability stabilization times, further changes are insignificant in both ethanol and DMF. The rejection also change over time. The R Remazol Brilliant Blue R decreases after 24 hours of filtration from 96 to 82% in the DMF, and from 71 to 58% in the ethanol.

  1. The authors take advantage of the metal ion chelating properties of the polymer to obtain OSN membranes. Similar approaches have been published already with various polymers by various groups inc. Jian Jin, Yan Wang, Szekely, Peinemann groups, which should be discussed, and the approaches and performance of these metal ion/polymer pairs compared in a final table before the Conclusions (i.e. Table 7).

The use of the chelating properties of metals in the fabrication of membranes is discussed before the final table (pages 11-12). The following articles have been cited: 10.1002/anie.201611927, 10.1021/acs.nanolett.5b00275, 10.1021/acsanm.2c01248, 10.1002/aic.17896, 10.1016/j.cej.2021.130941, 10.1039/D1TA02601A.

  1. The novelty of the research needs to be clarified in light of the similar research on alginate membranes for OSN that the authors published recently, one cited in the submitted manuscript but the other one is not mentioned (10.3390/membranes12020235; 10.4028/www.scientific.net/KEM.899.745).

The present manuscript is a continuation of our previous works. As you noticed, we have already published the article 10.3390/membranes12020235. The metal cations and the precipitant were varied in it in order to find the best conditions for the separation of aqueous and organic solutions. The article 10.3390/membranes12020235 considers a much narrower spectrum of metal cations (only calcium, aluminum and silver). The conclusion of article 10.3390/membranes12020235 contains the idea that alginate membranes are preferably crosslinked with aqueous salt solutions for the separation of organic solutions. We use this fact in the present manuscript. The process of preparation membranes is described in the experimental part. It was chosen precisely because of previous work. A link to this have been posted in the manuscript (page 4).

In another article (10.4028/www.scientific.net/KEM.899.745), we have already used all the crosslinking cations studied in this work: calcium, copper, zinc, iron, aluminum, and chromium. However, it rather demonstrated only the possibility of such an application. In the article (10.4028/www.scientific.net/KEM.899.745) you can find information about the stability of alginates in a wide range of organic solvents of various types (non-polar, polar protic, polar aprotic). Correlations between the physicochemical properties of solvents and their sorption into alginates have been demonstrated. Importantly, due attention was not paid to the filtering properties of the obtained membranes. In particular, the possibility of varying the filtration properties and the reasons why this becomes possible is not considered in the article.

In order to be correct and not to raise doubts about the novelty of the work, a discussion of this is included in the “introduction” section (page 3). In addition, the purpose of the work has been changed (page 3).

  1. Natural/green polymers for OSN are emerging and recent achievements should be mentioned (10.1016/j.memsci.2018.09.042; 10.1039/C8TA02697A; 10.1016/j.memsci.2022.120743; 10.1016/j.advmem.2022.100041).

In this manuscript the emphasis is not just on green membranes, but on those made from alginate. In this regard, references that you mentioned are inserted in the introduction (page 2), but without their detailed discussion.

  1. The degree of crosslinking should be determined and reported for all membranes studied.

Sodium alginate crosslinks with divalent and trivalent cations due to the ion exchange reaction. In this regard, the degree of crosslinking becomes possible to evaluate by the amount of sodium remaining after crosslinking. A line characterizing the concentration of sodium in the cross-linked samples was added to table 1. The sodium concentration is 0 in all cross-linked alginates. This allows us to conclude the degree of crosslinking is 100% for all alginates (page 7).

  1. Can the developed membranes be used for catalytic purposes? Some of the metal-polymer chelated systems are excellent catalysts. Is this a potential application area (catalytic membrane reactors) for these membranes?

This work is devoted to the use of alginate membranes for nanofiltration of organic solutions. The catalytic application of membranes is outside the scope of this study. It cannot be discussed in the manuscript so as not to mislead the reader.

This issue is interesting for further consideration and separate experimental studies. Thank you for drawing our attention to this possible application of the membranes we have developed.

At the moment, it is only possible to speculate on this topic theoretically. As we know, the use of catalytic membranes is based on their content of transition metal nanoparticles stabilized by polymers (doi.org/10.1016/j.matpr.2016.01.031). The role of the polymer is reduced to the ability to "retain" nanoparticles of metal catalysts and be permeable to solutions subjected to a catalytic reaction. It is reliably known alginate (doi.org/10.1016/j.actamat.2012.05.037; 10.1016/j.colsurfa.2013.12.008), like all other polysaccharides (doi:10.1016/j.jmrt.2020.02.097), is capable of reducing silver cations to nanoparticles (doi:10.1016/j.ijbiomac.2018.01.012). Alginate stabilize nanoparticles in the matrix (10.1016/j.radphyschem.2009.01.003), preventing their aggregation (doi.org/10.1016/j.carbpol.2016.05.018). At the same time, silver nanoparticles are known as catalysts (10.3390/polym8040105). This leads to the idea that silver alginate membranes theoretically can be used in catalytic membrane reactors. It is necessary to study their permeability to solutions subjected to a catalytic reaction. It is also good to study the selectivity of the catalytic reaction in such reactors and the stability of the process over time. Membranes for these purposes can be prepared by a similar method, which was described in the present manuscript.

  1. The authors propose an OSN cascade but it is out of the blue in the manuscript. Some discussions on the application of cascades in OSN should be added, the topic introduced. Separation performance should be evaluated experimentally or at least the rejections should be used for some cascade modeling to demonstrate the applicability and potential of the proposed cascade configuration.

The possibility of using cascades is indicated by differences in the rejection of negatively charged substances with different molecular weights. Imagine we have a mixture of Remazol Brilliant Blue R and Orange II dyes in the same solvent. Our task is to purify this solution. It is possible only with the use of chromium alginate with ROrangeII = 90% (table 5). However, this membrane is low permeable. Therefore, we suggest using a more highly permeable iron alginate-based membrane for pre-separation of Remazol Brilliant Blue R using.

This use of the cascade is currently only proposed in theory. It has not yet been tested experimentally. In this regard, we considered it necessary to remove this information from the manuscript. Perhaps this will be the topic of future research and subsequent publications.

Reviewer 3 Report

I have read the manuscript entitled, Membranes based on alginate salts with variable nanofiltration 2 properties’’. My comments and suggestions are mentioned below. Please consider during the revision process of your manuscript.

1.       Please add eco-friendly membranes in the title, to be more interesting by the readers.

2.       The authors must correct the following aspect, in my opinion you can't talk about figure 8 and then about figure 7 (please look at line 310-311).

3.      For each experimental parameter studied, a conclusion must be drawn in the section part of Results and Discussions.

4.       The authors must comment on the recycling efficiency of these membranes.

5.       Please writing the manuscript taking into consideration instructions of the Journal.

Author Response

Reviewer 3

Thanks for your review. The work has been corrected in accordance with your comments and remarks. All corrections in the text are highlighted in green.

  1. Please add eco-friendly membranes in the title, to be more interesting by the readers.

The new title of the article is as follows: "Eco-friendly OSN membranes for based on alginate salts with variable nanofiltration properties" (page 1). We have added in the name in accordance with your advice "Eco-friendly". In addition, it was clarified the membranes are used specifically for nanofiltration of organic solvents (OSN), and not for aqueous solutions.

  1. The authors must correct the following aspect, in my opinion you can't talk about figure 8 and then about figure 7 (please look at line 310-311).

Thank you. Corrected (page 10).

  1. For each experimental parameter studied, a conclusion must be drawn in the section part of Results and Discussions.

Micro-conclusions on the studied parameters are given at the end of sections 3.2 and 3.3 (pages 8, 11).

  1. The authors must comment on the recycling efficiency of these membranes.

Once the membranes have reached their useful life, the sodium alginate can be recovered for further use. Bivalent and trivalent salts of alginates are insoluble in water and organic solvents. However, they enter into ion exchange reactions with aqueous solutions of sodium bicarbonate.

2Alg-Ca + 2NaHCO3 = Ca(HCO3)2 + 2Alg-Na

This allows the polymer to be transferred to a soluble state for its further isolation from the solution by precipitation with acid solutions.

The strength of alginate binding by different cations differs from each other. In this regard, the dissolution of the films occurs at different concentrations of sodium bicarbonate (Table 2). The experiments were carried out using the same weight of cross-linked alginates.

Table 2 - solubility of alginate salts in sodium bicarbonate solutions

Men+

Fe3+

Ca2+

Zn2+

Al3+

Cr3+

Cu2+

ω NaHCO3

0.6%

1.2%

1.7%

4.3%

6.1%

6.4%

Thus, dissolution with aqueous solutions of sodium bicarbonate makes it possible to restore the polymer from membranes that have irreversibly lost their filtration properties.

  1. Please writing the manuscript taking into consideration instructions of the Journal.

Thank you. Corrected.

Round 2

Reviewer 1 Report

The manuscript was revised according to our comments, but there are still minor grammatical errors that need correction before the final acceptance. 

For example, 

Ln 38 “…….crosslinking is” change to ““…….crosslinking are”

Ln 52 add “such as” before “N,N-dimethyl formamide”, “and” before “N,N-dimethylacetamide”

Ln 99 Family name should only be cited in the text.

Ln 353 “a change of” to “a change in”

Ln 354 “also leads to a vary” to “also lead to a variation”

Ln 368 “The article [74] show” to “The article [74] shows”

The cases like this are so many! please reconsider all in the text.

Author Response

Thanks for the comments. The indicated errors have been corrected.

Reviewer 2 Report

The article is better, comments were taken care of. The title has a typo which needs to be corrected: delete the 'for' from the title:

Eco-friendly OSN membranes based on alginate salts with variable nanofiltration properties

Author Response

Thanks for the comments. All of these bugs have been fixed.

Reviewer 3 Report

 Accept in present form.

Author Response

Thanks to the reviewer for the recommendation. All previous comments were very helpful and allowed to improve the article.